# Adjusted trajectory of medication exposure taking into account the periodicity of dispensations and the number of dispensed packs and comparative analysis on EFEMERIS database

**Cécile Chouquet**[1☯*], **Anna-Belle Beau**[2‡], **Christine Damase-Michel**[2‡], **David Jeauneau**[1], **Isabelle Lacroix**[2‡], **Sabine Mercier**[3☯]

**1** Institut de Mathématiques de Toulouse, CNRS UMR 5219, Université Paul Sabatier Toulouse III, Toulouse, France, **2** Pharmacologie Médicale et Clinique, UMR INSERM 1295, Centre Hospitalier Universitaire, Faculté de médecine, Université Toulouse III, Toulouse, France, **3** Institut de Mathématiques de Toulouse, CNRS UMR 5219, Université Jean Jaurès Toulouse II, Toulouse, France

☯ These authors contributed equally to this work.
‡ ABB, CDM and IL also contributed equally to this work.
* cecile.chouquet@math.univ-toulouse.fr

**Data Availability Statement:** "The data relevant to this study cannot be shared publicly as it contains

## Abstract

We presented an adjustment method for the calculation of medication exposure trajectories based on the number of dispensed packs and the type of dispensations (occasional or regular). A comparative study based on the EFEMERIS data was carried out using three different scenarios of trajectory calculation depending on whether or not the number of packs and the periodicity of medication dispensations were taken into account. The impact of the scenario was highlighted using global indicators on the number of Define-Daily Dose (DDD) on all women exposed; the study of changes in individual trajectories from one scenario to another was carried out; we also compared the results of a clustering into four groups. If 65% of the trajectories remained unchanged, we could observe on the rest significant changes in number of DDD and/or on individual exposure profile. We observed 4% of trajectories that were attributed to a different cluster, and the clustering was of better quality with the adjustment method. Depending on the study context, an impact on cluster distribution could be observed for some maternal characteristics and neonatal outcomes. This was the case for a higher occurrence of neonatal pathology for neonates from mothers belonging to the cluster with high doses of psychotropics, thus reinforcing the conclusions of previous studies of a link between high exposure to psychotropic medications and presence of pathology for the newborn.

## Introduction

The challenge in measuring medication exposure lies in the different types of information available in health databases like dispensed medications, dispensing dates, number of

sensitive health data, including data about children, and potentially identifying information due to small sizes of some subpopulations. Additionally, the Commission Nationale de l'Informatique et des Libertés (CNIL) does not authorize the public sharing of the EFEMERIS data. Data access requires the submission of a research project according to a strict protocol to the cohort investigators and a request for individual access by VPN to the secure computing server. For more information, contact the Data Protection Officer (Dpo@chu-toulouse.fr) or the author Dr. Christine Damase-Michel (christine.damase-michel@univ-tlse3.fr), head of the "Medications, pregnancy and breastfeeding" information center, CERPOP, SPHERE team, INSERM, Toulouse."

**Funding:** The author(s) received no specific funding for this work.

**Competing interests:** The authors have declared that no competing interests exist.

dispensed packs, number of units per pack. A minimalist approach, and widely used so far, is to transform this information into exposed/non-exposed (possibly over time) [1, 2]. This binary approach leads to the quantitative and temporal aspect of the medication load being missing. Other approaches presented in [3] have recently been proposed to model drug exposure, by considering some important dimensions of treatment such as dosage, duration and timing of exposure and to identify exposure profiles by using unsupervised clustering methods such as Group-based trajectory methods [4–6] or Kmeans for longitudinal data [7, 8]. In [7], a new pharmacoepidemiological method has also been developed to take into account intensity and evolution of medication exposure and to estimate medication load in the form of an exposure trajectory over time. This method has been applied to pregnant women exposed to psychotropic medications from the EFEMERIS database, a cohort of pregnant women and their outcomes in South-West France [9]. We proposed to expand this method using another mode of calculating the medication load, which takes into account both the periodicity of medication dispensations and the number of dispensed packs.

We first recall the initial method and then describe the one developed. To assess the numerical contribution of the new calculation method versus the initial method, a comparative analysis was performed based on three scenarios, depending on whether or not the periodicity and the number of dispensed packs were taken into account. The main changes due to the proposed method on the medication exposure distribution and on exposure clusters are detailed in Section Results and discussion.

## Materials and methods

### Initial method of calculating exposure trajectories

To quantify exposure to all dispensed medications, physical quantities of medications in number of units were transformed into a standard unit of measurement proposed by the World Health Organization: the Defined-Daily Dose, denoted by DDD [10]. Women taking psychotropic medications are often exposed to several ones (for example, antidepressant and anxiolytics) and the use of DDDs allows to aggregate units of different medications, in our case the different types of psychotropic medications (identified using the ATC code). Exposure periods were computed from dispensation dates and number of units per pack. The number of dispensed packs was not available and set at 1 by default. In a general manner, it is assumed that for each dispensed medication, women are exposed to one DDD per day [7]. For example, for one dispensed pack of 7 DDD, the treatment duration was estimated at 7 days with one DDD per day. If 14 DDDs were dispensed, the treatment duration was estimated at 14 days. Thus, in the initial mode of calculation, the treatment periods of the same active substance were placed one after the other. In case of pregnancy, some DDDs could be shifted after birth and were therefore not taken into account in the calculation of the exposure measurement. This computation could be apply to a medication, or several medication of the same class: after superimposing individual periods of exposure, the number of DDDs per day and per medication was calculated by cumulating for all psychotropic medications dispensed on a given day of a woman's pregnancy. A sequence of daily quantitative measurements of psychotropic exposure was therefore available for each woman, which could be called daily exposure trajectory.

### Proposed method of calculating exposure trajectories based on periodicity adjustment

In case where the number of dispensed packs was available, the first step was to calculate the total number of units dispensed at each dispensation of a specific medication in order to

estimate the active medication load closer to reality. Then, the periodicity was included by differentiating single or occasional dispensing from regular dispensing. A dispensation was considered as single or occasional if it was not renewed over a period of 45 days. In this case, the distribution of the corresponding total DDDs was done as for to the initial method, ie. put one after the other with one DDD per day. Dispensations were defined as regular if they were repeated in a period of less than 45 days, based on French dispensing habits for one month of treatment, adding a flexibility of 15 days. If a specific medication was dispensed regularly for a woman, we calculated the average number of DDDs dispensed over all regular dispensations ($\overline{N}_{DDD}$) and the average duration between two regular dispensations ($\overline{D}_{between}$). We deduced the ratio $F = \overline{N}_{DDD}/\overline{D}_{between}$ which estimated, for the considered pregnancy, the number of DDDs per day distributed between the first and the last regular dispensation. From the last dispensation, the remaining number of DDDs per day was $F/2$ until exhausted to consider possible weaning.

As the initial method, we then deduced the total number of DDDs per day by cumulating all medications of interest (psychotropic) dispensed on a given day of a woman's pregnancy. In this study, we also cumulated the exposure measurement by week, to obtain weekly exposure trajectory for each exposed pregnancy.

This method to construct longitudinal trajectories of drug exposure required data containing the following information: name and ATC of the drug dispensed from which the dosage, form and number of days of treatment per pack are deduced; the number of packs dispensed; the date of dispensing. Such data are commonly available in European Health care data sources [11]. We proposed an example of application on the EFEMERIS database presented in the next subsection.

## Data

Data for our analysis comes from the EFEMERIS database [9]. The EFEMERIS cohort was approved by the French Data Protection Authority on 7 April 2005 (authorization number 05-1140). This study was performed on anonymized patient data. The women included in the EFEMERIS database were informed of their inclusion and that their collected and anonymized data can be used for medical research purposes and can thus be published. They could oppose the use of their data at any time. The study was approved by the EFEMERIS steering group. Data were handled and stored in accordance with the General Data Protection Regulation. The EFEMERIS database contains: data about medication dispensed to pregnant women (from CPAM, the national medical insurance organisation) in Haute-Garonne; data concerning pregnancy outcomes; data on the children obtained from mother-and-child health services (PMI, "Protection Maternelle et Infantile") which records information about the child and delivers the mandatory health certificates for children at eight days, nine months and 24 months of age. Note that drug prescription and dispensation for pregnant women of the EFEMERIS database are representative of the general French population [12]. We also have information on pregnant women such as age, presence of diabetes or preeclampsia. The present study was restricted to mother-child pairs included in EFEMERIS between June 1, 2011 and December 31, 2020. From data extraction performed on April 11, 2023, we retrieved over this period 24,138 psychotropic medication dispensations among 6,820 live pregnancy outcomes for which the presence or absence of neonatal pathology was known. We considered in our study the pair, pregnancy-outcome of pregnancy, which will be noted pregnant woman or pregnancy in the following document in the absence of ambiguity. In this context, for each pregnancy, the exposure trajectory described the exposure to psychotropic medications. More precisely, the exposure trajectories started from 4 weeks before pregnancy to take into account

the potential long-term presence of medications in the woman's body and its potential effect on the fetus from the start of pregnancy. To calculate such an exposure trajectory from 4 weeks before pregnancy, medication dispensations from week -13 were used, because a treatment taken at -4 have been dispensed between -13 and -4. But DDD before -4 were not considered for analysis of exposure trajectories. Weekly measures of exposure to psychotropic medications were calculated for each pregnant woman exposed at least once between -4 weeks before pregnancy and the end of pregnancy, called pregnancy period in the rest of this document.

## Comparative analysis approach

We considered the three following scenarios for our comparative study to assess the contribution of the proposed method:

- Scenario 1: As if the number of dispensed units were not available (set by default at 1) and no periodicity adjustment,

- Scenario 2: Available number of dispensed units and no periodicity adjustment,

- Scenario 3: Available number of dispensed units and periodicity adjustment.

We first assessed the change of number of exposed pregnancies due to the number of packs dispensed during week -13 and -5 and the periodicity adjustment. For each scenario, we then computed some global indicators on the whole set of the trajectories of exposed pregnancies, gathered together (Table 1). The sum of DDDs during the pregnancy period was calculated to evaluate the change of the DDD number between scenarios. As the number of exposed pregnancies changed between scenarios, we also computed the global average DDD per week, the standard deviation, the 90 and 95 percentile, and the maximum. In a second step, the individual trajectories for each pregnancy were computed and compared using the three scenarios. The percentage of pregnancies without any change of trajectories between scenarios was calculated. When a change occurred, the variation from one scenario to another of the total of DDDs was quantified and analyzed. In a last step, we evaluated the impact using the periodicity adjustment on the creation of exposure clusters. As in [7, 13], we used the Kmeans method for longitudinal data [14], an unsupervised clustering method based on an algorithm for which the aim is to partition the observations (here the exposure trajectories) into a given number of clusters. We first studied concordance between those two partitions deduced from scenario 2 and scenario 3. Secondly, we compared clusters from scenario 2 and 3 in terms of average weekly DDD values, maternal characteristics and neonatal outcomes.

## Results and discussion

In our studied dataset, we observed that 28% of the psychotropic exposed pregnancies had more than one pack of psychotropic medication dispensed at least once. By reasoning on the number of dispensing, this concerned 32% of all the dispensing.

**Table 1. Indicators of comparison of DDD values between the three scenarios.**

|  | Number of exposed pregnancies at least one week between -4 and 38 | Sum of DDDs between -4 and 38 | Distribution of weekly DDD | | | |
|---|---|---|---|---|---|---|
|  |  |  | Mean (sd) | 90 percentile | 95 percentile | max |
| Scenario 1 | 5467 | 258407 | 1.13 (3.00) | 6.5 | 7.0 | 47 |
| Scenario 2 | 5561 | 341636 | 1.47 (3.60) | 7.0 | 7.0 | 50 |
| Scenario 3 | 5590 | 366438 | 1.57 (4.06) | 6.5 | 8.7 | 84 |

## Step 1: Comparison on DDD values distribution (Table 1)

Remember that the trajectories were calculated from dispensing from 13 weeks before pregnancy. This could impact whether a pregnancy was classified as exposed from 4 weeks before pregnancy and thus could change the exposure trajectory. Taking into account the number of dispensed units (from scenario 1 to 2) allowed to recover 94 exposed pregnancies, and adding the periodicity adjustment (from scenario 2 to 3) allowed to recover 29 more exposed pregnancies, ie. an increase of 2.2% in the number of pregnancies exposed at least one week between -4 and 38 weeks of pregnancy (second column of Table 1). The sum of DDDs over all the trajectories increased by 32% from scenario 1 to 2 and by 42% from scenario 1 to 3 across global trajectories of all exposed pregnancies (third column of Table 1). Considering only the impact of the periodicity adjustment (from scenario 2 to 3), the sum of DDDs increased globally by 7%.

Note that the increase in the sum of DDD between scenarios 1 and 2 was due to the fact that there were more pregnancies classified as exposed and that exposure measures were logically increased if the number of units is greater than 1. The increase from scenario 2 to 3 was also explained by the fact that in scenario 2, some DDDs were shifted after the end of pregnancy, while with scenario 3, the periodicity adjustment allowed them to be positioned more appropriately during pregnancy period. The average sum of DDD per pregnancy increased for example from 47.3 for scenario 1 to 65.6 for scenario 3, ie. almost 20 DDD more over the entire pregnancy period (from -4 to around 38 weeks). We also studied the weekly DDD distribution (four last columns of Table 1). The periodicity adjustment had the effect of largely increasing the maximum DDDs as well as the largest values (see the 95 percentiles). Note that the average increased slightly, due the fact that at least 75% of the DDDs were equal to 0 for each of the three scenarios.

## Step 2: Impact on individual trajectories

Step 2 considered the pairwise differences between the three trajectories of each exposed pregnancy. From scenario 1 to 2 (adding the number of dispensed units), 68% of pregnancies had a different trajectory, ie. at least one week with a different DDD, which was consistent with the percentage of dispenses of only one pack. Only 10% of trajectories increased on average by at least one DDD per week, and up to 10 DDD per week.

From scenarios 2 to 3 (adding the periodicity adjustment), 65% of the trajectories were strictly identical. Among the 35% of pregnancies having different trajectories, one third had a total of DDDs decreased with a maximum drop corresponding to an average of 3 DDDs per week. This decrease could be explained mainly by the modified calculation of DDD before -4 weeks which could lead to a greater number of DDDs before week -4 and therefore lower after. Half of the modified trajectories realized an increase of the total DDDs corresponding to 1.3 DDDs per week on average with a standard deviation equal to 2.1. For the rest of these modified trajectories, the total of DDDs did not change due to a compensation effect between weeks. We observed 2% of all trajectories (ie. 106 trajectories) associated with an increase of the total of DDD of between approximately 2 and 20 DDD per week on average.

Fig 1 represents examples of trajectories calculated according to the 3 scenarios for 4 exposed pregnancies. For these four pregnancies, the number of dispensations was between 24 and 39 corresponding to between 2 and 7 different dispensed ATC, with half of dispensations of more than one pack. In the three first examples, we observed that taking into account the number of packs was mainly enhanced by the periodicity adjustment. This revealed changes in the trajectory profiles: a higher exposure at the start of pregnancy, followed by a gradual decrease for pregnancy n°1, a higher exposure throughout the pregnancy period for pregnancy

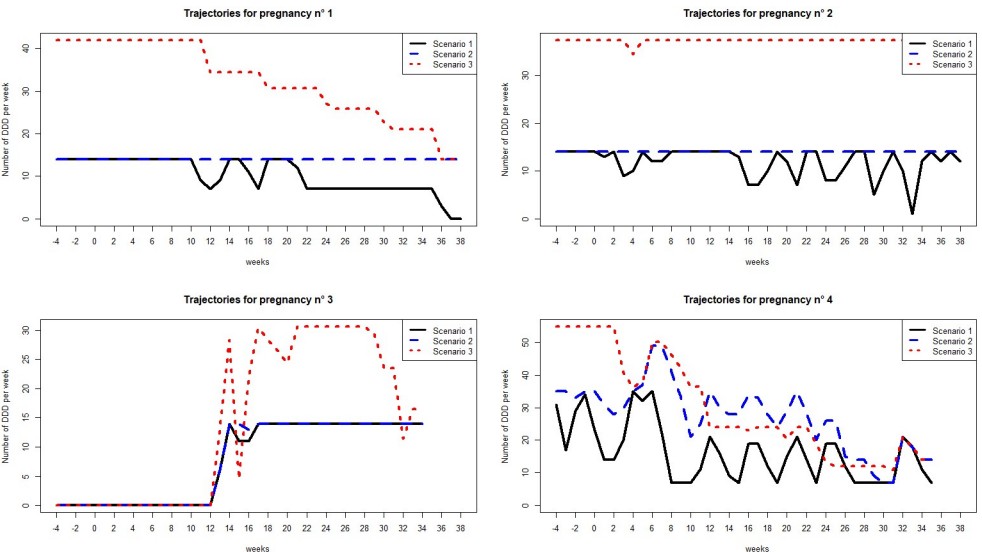

**Fig 1. Examples of individual exposure trajectories.** Examples of exposure trajectories calculated according to three calculation scenarios, for four exposed pregnancies. The woman of pregnancy n˚1 received 24 medication dispensations of 3 different ATC over the pregnancy period, of which 20 with more than one pack. The woman of pregnancy n˚2 received 36 medication dispensations of 2 different ATC over the pregnancy period, of which 32 with more than one pack. The woman of pregnancy n˚3 received 25 medication dispensations of 2 different ATC over the pregnancy period, of which 13 with more than one pack. The woman of pregnancy n˚4 received 39 medication dispensations of 7 different ATC over the pregnancy period, of which 19 with more than one pack.

n˚2 and a medication load multiplied by 2 after the first trimester for pregnancy n˚3. The changes were less significant for the pregnancy n˚4.

## Step 3: Impact on clustering

Using the individual weekly exposure trajectories calculated with scenarios 2 and 3, we implemented the Kmeans method for longitudinal data to create exposure clusters. We evaluated the impact of using the periodicity adjustment on the creation of exposure clusters by comparing the partitions of scenarios 2 and 3. The quality of the clustering measured by the Calinski-Harabatz criterion was better for scenario 3: 2302 for scenario 3 versus 2010 for scenario 2. Based on four clusters [7], it could be seen that taking into account the periodicity adjustment did not change the representative trajectory profile of the four clusters (see average trajectories in Fig 2). Note that if the representative profile of cluster D (strong exposure throughout pregnancy) does not change, it is nevertheless positioned higher in Scenario 3, which will be detailed in Table 3.

**Concordance between scenarios 2 and 3.** The concordance study between the two partitions (see Table 2) showed two types of changes:

- 45 pregnancies changed status from exposed to non-exposed or vice versa, with 37 pregnancies non-exposed with scenario 2 that became exposed (cluster A) with scenario 3, and 8 exposed with scenario 2 that lost the status of exposed pregnancy (cluster A) with scenario 3;

- among the 5553 pregnancies exposed in both scenarios, 225 (4%) switched exposure cluster: 79 pregnancies moved to one higher cluster and 109 to one lower cluster; 37 pregnancies changed in at least two exposure clusters level, with 33 from cluster C to cluster A, and 4 from A to C.

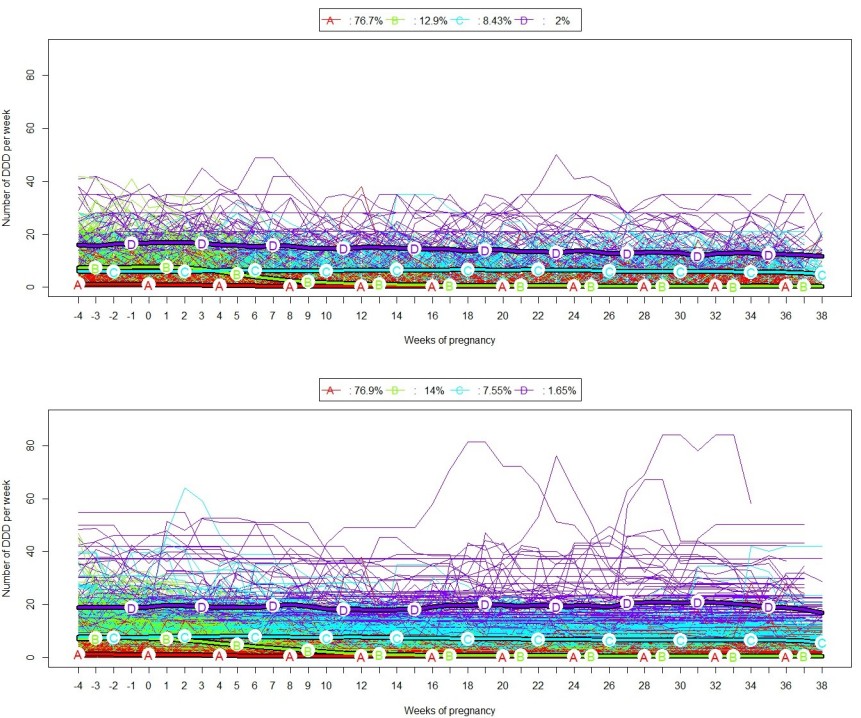

**Fig 2. Individual exposure trajectories and average trajectories for clusters.** Four clusters were identified by K-means method for longitudinal data. Individual trajectories in the same cluster are plotted in the same color. The top figure represents the exposure trajectories calculated according to scenario 2, and the bottom one according to scenario 3. The average trajectories of each cluster are represented by the bold lines and designated by letters.

In summary, among this total of 270 changes, one third (82 trajectories) could be qualified as major, considering pregnancies non-exposed to one of the two scenarios and exposed to the other, or a change of at least two levels of exposure clusters.

**Comparison of weekly DDD values and neonatal pathology by cluster between scenarios 2 and 3.** We gathered in Table 3 indicator values summarizing the trajectory of each exposed pregnancy. Here, we calculated, for each pregnancy, its average weekly DDD. We then deduced mean, standard deviation (sd), median and range interval over all the trajectories. Note that Table 1 provided the overall average of all weekly DDD values combined, without considering the grouping into trajectories. In contrast, Table 3 (Column 3) showed the

**Table 2. Concordance table between scenarios 2 and 3.**

| | | Scenario 3 | | | | | Sum |
|---|---|---|---|---|---|---|---|
| | | A | B | C | D | Non-exposed | |
| Scenario 2 | A | 4198 | 54 | 4 | 0 | 8 | 4264 |
| | B | 28 | 684 | 5 | 0 | 0 | 717 |
| | C | 33 | 42 | 374 | 20 | 0 | 469 |
| | D | 0 | 0 | 39 | 72 | 0 | 111 |
| | Non-exposed | 37 | 0 | 0 | 0 | 80909 | 80949 |
| | Sum | 4296 | 780 | 422 | 92 | 80917 | 86507 |

**Table 3. Indicators of comparison of average weekly DDD values and neonatal pathology by cluster between scenarios 2 and 3.**

| | Exposure clusters | | | | All exposed pregnancies |
|---|---|---|---|---|---|
| | **A** | **B** | **C** | **D** | |
| Scenario 2 | | | | | |
| Nb of pregnancies | 4264 | 717 | 469 | 111 | 5561 |
| Average weekly DDD | | | | | |
| Mean (SD) | 0.55 (0.56) | 2.63 (1.41) | 6.92 (1.96) | 16.3 (4.61) | 1.67 (2.99) |
| Median [min-max] | 0.37 [0.01,3.68] | 2.22 [1.11,10.3] | 6.83 [3.75,12.7] | 15.3 [11.1,31.3] | 0.51 [0.01,31.3] |
| Neonatal pathology | | | | | |
| n (%) | 465 (10.9%) | 81 (11.3%) | 74 (15.8%) | 24 (21.6%) | 644 (11.6%) |
| Scenario 3 | | | | | |
| Nb of pregnancies | 4296 | 780 | 422 | 92 | 5590 |
| Average weekly DDD | | | | | |
| Mean (SD) | 0.58 (0.62) | 2.66 (134) | 8.13 (2.64) | 21.9 (7.27) | 1.79 (3.59) |
| Median [min-max] | 0.37 [0.01,4.42] | 2.27 [1.03,9.58] | 7.50 [4.16,15.1] | 19.3 [14.6,53.8] | 0.53 [0.01,53.8] |
| Neonatal pathology | | | | | |
| n (%) | 474 (11.0%) | 85 (10.9%) | 65 (15.4%) | 24 (26.1%) | 648 (11.6%) |

average of the trajectory-specific averages, which could differ from the overall average if the trajectories vary in size. These represented two distinct approaches to summarizing the data.

The comparison of the two scenarios showed that:

- Lower exposure clusters (A and B) had larger populations in scenario 3 compared to scenario 2. However, the indicators remained unchanged, likely due to a mass effect resulting from the larger population sizes.

- Higher exposure groups had smaller populations, higher mean and variability of weekly DDDs in Scenario 3 compared to Scenario 2. More specifically for cluster D, the average weekly exposure per woman increases by about one standard deviation from 16.3 (scenario 2) to 21.9 (scenario 3), corresponding to an additional 5 DDDs per week on average. The variability of average weekly exposure for trajectories of cluster D also increased from 4.61 (scenario 2) to 7.27 (scenario 3), mainly due to even higher extreme values.

We also studied in Table 3 the impact of these clustering changes on neonatal pathology: the proportion of neonatal pathology in cluster D increased with the periodicity adjustment from 21.6% (out of 111 pregnancies) to 26% (out of 92 pregnancies), whereas we have observed that the proportion of premature births changed very little with 9% in scenario 2 and 9.8% in scenario 3.

Concerning variables affecting the mother, it was observed for example that the proportion of pre-eclampsia per cluster remains unchanged from one scenario to another, but there was a higher proportion of diabetes reported among pregnancies in the high exposure cluster in scenario 3 (19.5%) as compared with scenario 2 (15%).

## Conclusion

In the case where the number of dispensed packs was available, we highlighted an impact of adjusting the calculation of exposure measure by taking into account the periodicity of dispensations. The analyzes on the total number of DDDs over all trajectories or on individual ones showed that the method increased the average exposure measure quite slightly. A percentage of 65% of the exposed pregnancies did not have any change in the weekly individual

trajectories using the different methods. Among the changed trajectories, 90% showed an increase or a decrease of the cumulative medication exposure, and the remaining 10% had changes according to the weeks that are compensated. There could be a great variability in changes in the medication exposure and thus for a non-negligible number of trajectories, an increased number of DDDs and a modified exposure trajectory profile could be observed. The adjusted trajectories led to clusters with no radical modifications but with better quality and higher level of exposure for the women most exposed. Considering others variables, adjustment variables or outcomes in pharmacoepidemiology studies, their distribution by cluster could be impacted or not.

## Acknowledgments

We thank Caroline Hurault-Delarue who provided the foundation for the work in her Phd, and Anthony Caillet for the data extraction.

## Author Contributions

**Conceptualization:** Cécile Chouquet, Sabine Mercier.

**Formal analysis:** Cécile Chouquet, David Jeauneau, Sabine Mercier.

**Investigation:** Cécile Chouquet, Sabine Mercier.

**Methodology:** Cécile Chouquet, Anna-Belle Beau, Christine Damase-Michel, David Jeauneau, Isabelle Lacroix, Sabine Mercier.

**Project administration:** Cécile Chouquet, Sabine Mercier.

**Resources:** Christine Damase-Michel, Isabelle Lacroix.

**Software:** Cécile Chouquet, David Jeauneau, Sabine Mercier.

**Supervision:** Cécile Chouquet, Christine Damase-Michel, Isabelle Lacroix, Sabine Mercier.

**Validation:** Cécile Chouquet, Anna-Belle Beau, Christine Damase-Michel, Isabelle Lacroix, Sabine Mercier.

**Visualization:** Cécile Chouquet, Sabine Mercier.

**Writing – original draft:** Cécile Chouquet, Anna-Belle Beau, Christine Damase-Michel, Isabelle Lacroix, Sabine Mercier.

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
