## [Decision Letter · Decision Letter 0]

22 Oct 2024

PONE-D-24-29477Adjusted trajectory of medication exposure taking into account the periodicity of dispensations and the number of dispensed packs and comparative analysis on EFEMERIS databasePLOS ONE

Dear Dr. Chouquet,

Thank you for submitting your manuscript to PLOS ONE. After careful consideration, we feel that it has merit but does not fully meet PLOS ONE’s publication criteria as it currently stands. Therefore, we invite you to submit a revised version of the manuscript that addresses the points raised during the review process.

We look forward to receiving your revised manuscript.

Kind regards,

Arianit Jakupi, PhD

Academic Editor

PLOS ONE

Journal Requirements:

3. Please note that your Data Availability Statement is currently missing [the repository name and/or the DOI/accession number of each dataset OR a direct link to access each database]. If your manuscript is accepted for publication, you will be asked to provide these details on a very short timeline. We therefore suggest that you provide this information now, though we will not hold up the peer review process if you are unable.

Reviewers' comments:

Reviewer's Responses to Questions

**Comments to the Author**

1. Is the manuscript technically sound, and do the data support the conclusions?

Reviewer #1: Yes

Reviewer #2: Yes

2. Has the statistical analysis been performed appropriately and rigorously? 

Reviewer #1: Yes

Reviewer #2: Yes

3. Have the authors made all data underlying the findings in their manuscript fully available?

Reviewer #1: Yes

Reviewer #2: No

4. Is the manuscript presented in an intelligible fashion and written in standard English?

Reviewer #1: Yes

Reviewer #2: Yes

5. Review Comments to the Author

Reviewer #1: The manuscript entitled ‘Adjusted trajectory of medication exposure taking into account the periodicity of dispensations and the number of dispensed packs and comparative analysis on EFEMERIS database’ addressed a pharmacoepidemiologic method for the calculation of psychotropic medication exposure trajectories on the basis of the number of dispensed packs and the type of dispensations.

The manuscript is scientifically sound, composed according the scientific methodology for writing the scientific papers and represents a valuable contribution for scientists involved in the pharmacoepidemiology studies.

In the introduction section authors concisely provided the background information of the study topic and explained the principal aims of the study. In materials and methods sections authors adequately described initial and proposed method of calculating exposure trajectories, data collection, handling, storing, and analysis.

The results of study are presented in tabulated and graphical form, in a proper format and easy to understand and interpret. The study results are adequately discussed.

Conclusions are supported by the data, summarizing the results of this study and indicating the most significant findings and their importance. References are properly cited.

Comment 1: Although throughout results and discussion section, results are adequately discussed, authors did not provide sufficient number of references of recent studies to support and explain the obtained data.

Comment 2. Isabelle Lacroix appears as a very high percentage in the list of references. We suggest that this percentage be reduced so that it is not considered excessive self-citation.

Reviewer #2: The manuscript presents a technically sound study with rigorous methodology, particularly in comparing three different scenarios of medication exposure calculation. The data analysis, including the use of clustering methods and statistical tests, is appropriate and supports the conclusions drawn.

The statistical analysis is performed rigorously, and the results are valid. The use of the EFEMERIS database provides a strong dataset, though the data cannot be publicly shared due to privacy restrictions. The authors have clearly explained the process for data access, adhering to ethical standards.

The manuscript is well-written, clear, and free from major grammatical or typographical errors. The scientific terminology is appropriate for the audience.

Overall, the study is a valuable contribution to the field of pharmacoepidemiology, with minor areas for clarification, such as the generalizability of the findings. There are no concerns about research ethics or dual publication.

Comments and remarks:

The data come from a specific area (Haute-Garonne, France), and it would be helpful for the authors to address more broadly the potential limitations of generalizing the results to other populations.

The discussion on the impact of classification changes shows that only 4% of trajectories change clusters, so a more in-depth discussion of these changes and their impact on the results could be useful.

6. PLOS authors have the option to publish the peer review history of their article (what does this mean?). If published, this will include your full peer review and any attached files.

Reviewer #1: No

Reviewer #2: No

---

## [Author Response · Author response to Decision Letter 0]

16 Dec 2024

The text below corresponds to the file "Response to Reviewers" (see in attached file). 

Dear Editor and dear reviewers:

We wish to submit our revised manuscript of our research article for publication in PLOS ONE, titled “Adjusted trajectory of medication exposure taking into account the periodicity of dispensations and the number of dispensed packs and comparative analysis on EFEMERIS database” (coauthored by Cécile Chouquet, Anna-Belle Beau, Christine Damase-Michel, David Jeauneau, Isabelle Lacroix and Sabine Mercier).

First of all, we thank you for your answers and your relevant remarks allowing us to clarify and improve the manuscript. Below we respond to the comments of the academic editor and the reviewers on each point requiring a response after recalling them in italics.

Answers to the academic editor

Point 2. We note that you have indicated that there are restrictions to data sharing for this study. For studies involving human research participant data or other sensitive data, we encourage authors to share de-identified or anonymized data. However, when data cannot be publicly shared for ethical reasons, we allow authors to make their data sets available upon request.

Answer: Concerning data sharing: data cannot be publicly shared for ethical and legal restrictions. Indeed EFEMERIS data is pseudonymized. Data are not directly identifying but are not appropriate to share because:

- It concerns sensitive health data (medication exposure, and children pathologies).

- In some analyses, the number of pregnant women is less than 5, which makes the data potentially identifiable.

- The Commission Nationale de l'Informatique et des Libertés (CNIL) does not authorize us to share EFEMERIS data. 

To confirm the non-authorization of access to data, the contact is the Data protection officer of the CHU of Toulouse (Dpo@chu-toulouse.fr). 

Point 3. Please note that your Data Availability Statement is currently missing [the repository name and/or the DOI/accession number of each dataset OR a direct link to access each database]. If your manuscript is accepted for publication, you will be asked to provide these details on a very short timeline. We therefore suggest that you provide this information now, though we will not hold up the peer review process if you are unable.

Answer: Given the restrictions on sharing data from the EFEMERIS cohort, point 3 does not concern us.

Point 4. Please review your reference list to ensure that it is complete and correct. If you have cited papers that have been retracted, please include the rationale for doing so in the manuscript text, or remove these references and replace them with relevant current references. Any changes to the reference list should be mentioned in the rebuttal letter that accompanies your revised manuscript. If you need to cite a retracted article, indicate the article’s retracted status in the References list and also include a citation and full reference for the retraction notice.

Answer: The list of references has been checked, reviewed and expanded. We have also made some modifications to address the comments 1 and 2 of reviewer 1 (see below). 

Answers to the review Comments

Reviewer 1

Comment 1. Although throughout results and discussion section, results are adequately discussed, authors did not provide sufficient number of references of recent studies to support and explain the obtained data.

Comment 2. Isabelle Lacroix appears as a very high percentage in the list of references. We suggest that this percentage be reduced so that it is not considered excessive self-citation.

Answer to comments 1 and 2: 

The Section Introduction has been revised to take into account additional references on methods recently developed and/or used to model drug exposure over time and to group trajectories into exposure profiles (cf. ref. [4-8]). Furthermore, we kept only one of the 2 references presenting EFEMERIS in Section Data to balance the number of references between those attributable to the authors of our article, and those of other research teams. In the same section, we added one reference on the representativeness of the EFEMERIS database at the national level (cf. added ref. [12]). The number of references has therefore increased from 7 to 14 references, including 4 citing one of the authors of our article (compared to 5 out of 7 in the previous version).

Reviewer 2

Point 1: The data come from a specific area (Haute-Garonne, France), and it would be helpful for the authors to address more broadly the potential limitations of generalizing the results to other populations. 

Answer: We responded to the remarks about the generalizability of the findings from two points of view: one concerning the representativeness of EFEMERIS data, and the other the generalization of the method with regard to the information provided in the databases.

- The first concerns the EFEMERIS database whose women studied come from Haute-Garonne. A study of the representativeness of this database was carried out by Demailly {\\it et al.} (cf. added ref. [12]) on the prescription and dispensing data of drugs in pregnant women. We added this element in the Section Data (lines 90-91) and cited this reference. 

- We also responded to your comment from an angle relating to the generalization of the method itself and its use on other databases. For this purpose, we have added a paragraph (lines 70-75) at the end of the Subsection “Proposed method … periodicity adjustment” to mention the necessary information for the application of the method: namely the name and ATC of the dispensed drug, the number of dispensed packs, and the date of dispensing. Such data are commonly available in European Health care data sources as mentioned by Charlton (cf. added ref. [11]). 

The purpose of the article is to present the method. The EFEMERIS database is used as example to illustrate the method. The pharmaco-epidemiological developed results themselves will be the subject of other articles currently being written.

Point 2: Discussion on the impact of classification changes shows that only 4% of trajectories change clusters, so a more in-depth discussion of these changes and their impact on the results could be useful.

Answer: Step 3 of Section Results and discussion has been totally modified. We integrated two new tables. 

- The first one (Table 2) corresponds to a concordance table between the clusters of scenarios 2 and 3 (see Subsubsection “Concordance between scenarios 2 and 3”) to clarify and better understand the changes in partition. We commented on the main changes in numbers of trajectories by cluster. We then highlighted what we call major changes meaning, non-exposed trajectory becoming exposed or vice versa, or trajectories changing in at least two exposure cluster levels. 

- The second one (Table 3) presents in Subsubsection “Comparison of weekly DDD values and neonatal pathology by cluster between scenarios 2 and 3”, classical statistical indicators for each cluster and each scenario to see how the content of the four clusters changes. We discuss the main changes in particular for the cluster D and the impact on the neonatal pathology. 

More targeted analyses to model neonatal risk of disease based on exposure clusters (such as logistic regression or other) will be the subject of other publications. 

All changes (those mentioned above and other more minor ones) made to the original version were highlighted in the marked-up copy of our manuscript. Please note that we have also modified and completed the Human Subjects Research Checklist. 

We thank you again for your feedback and hope to have responded fully.

Sincerely,

---

## [Editor Report · Decision Letter 1]

22 Dec 2024

Adjusted trajectory of medication exposure taking into account the periodicity of dispensations and the number of dispensed packs and comparative analysis on EFEMERIS database

PONE-D-24-29477R1

Dear Dr. Chouquet,

We’re pleased to inform you that your manuscript has been judged scientifically suitable for publication and will be formally accepted for publication once it meets all outstanding technical requirements.

Kind regards,

Arianit Jakupi, PhD

Academic Editor

PLOS ONE
---

## [Editor Report · Acceptance letter]

13 Jan 2025

PONE-D-24-29477R1 

PLOS ONE

Dear Dr. Chouquet, 

I'm pleased to inform you that your manuscript has been deemed suitable for publication in PLOS ONE. Congratulations! Your manuscript is now being handed over to our production team.

Kind regards, 

on behalf of

Dr Arianit Jakupi 

Academic Editor

PLOS ONE